# Applicability of Variable-Rate Nitrogen Top Dressing Based on Measurement of the Within-Field Variability of Soil Nutrients for Cabbage Production

Yuka Nakano [1,*], Shintaro Noda [1], Yasunari Miyake [1], Masayuki Kogoshi [1], Fumio Sato [2] and Wataru Iijima [1]

1  Research Center for Agricultural Robotics, National Agriculture and Food Research Organization (NARO), Tsukuba 305-0856, Japan
2  Institute of Vegetable and Floriculture Science, National Agriculture and Food Research Organization (NARO), Tsukuba 305-8519, Japan
*  Correspondence: yuka88@affrc.go.jp

**Abstract:** To improve the efficiency of nitrogen (N) fertilization, it is necessary to perform rapid direct measurements in the field rather than time-consuming laboratory-based chemical analysis. Herein, crop and soil data from the early stages of cabbage growth were acquired through two fall cultivations. Chlorophyll meter value, height, and projected leaf area were evaluated as crop indicators. A positive correlation was observed between the projected leaf area or its rate of increase 2–3 weeks after transplantation and head fresh weight (FW). After comparing two water-content reflectometers (WCR) and a nitrate sensor, we selected a WCR with a 12 cm-long rod as the soil indicator. The diagnostic method was verified using varying amounts of N basal fertilizer during spring cultivation. The variable rate of N top dressing (25, 50, and 75% total N) based on the electrical conductivity (EC) 14 days after transplantation reduced the subsequent EC variability. No differences in head FW were observed between the treatments. A 25% reduction in N fertilizer was considered possible for half of the plots. The quantity of inorganic N extracted by potassium chloride from the crop soil after cultivation was unaffected by the amount of N fertilizer. Therefore, the diagnostic method proposed herein is suitable for soil N management.

**Keywords:** *Brassica oleracea* L. var. *capitata*; fertilizers; precision farming; soil sensing; water-content reflectometer

## 1. Introduction

In Japan, the need for convenience foods is increasing annually, as households are purchasing cut vegetables, frozen vegetables, and pre-prepared vegetable side dishes more frequently [1]. Moreover, the demand for vegetables for processing and commercial use is expected to remain high. Producers must manage cultivation according to their contracts with clients. Soil management, especially nitrogen (N) fertilization, is one of the most important aspects of open-field vegetable cultivation. International fertilizer prices have increased markedly since 2021, and the Japanese government is rushing to mitigate the impact on farmers. Thus, there is a growing need for technology to control the application of chemicals to farmlands. The Japanese government's Strategy for Sustainable Food Systems—MeaDRI—aims to enhance potential production while ensuring sustainability. To this effect, it has set a goal of reducing chemical fertilizer use by 30% by 2050 [2].

Cabbage is a major open-field vegetable crop with a high N demand [3]. Growers usually decide on the amount of N fertilizer to be applied based on their experience and the fertilizer application standards set by prefectures in Japan. Fertilizer standards of the primary cabbage-producing regions require an N application rate of 28–30 g N m$^{-2}$ in Aichi [4] and 20–23 g N m$^{-2}$ in Gunma [5]. Cabbage absorbs N most actively around the early head formation stage and more slowly in the head fill stage [6]. Therefore, as

guided by the Aichi prefectural government [4], in addition to basal fertilizer, one or two top dressing fertilizers are applied before the head formation stage begins, at the same time as weeding between rows. Fertilizer types and application methods have been tested to reduce the cost and environmental impact of cabbage production. For example, slow-release fertilizers [7,8], localized dressings [3,9,10], organic fertilizers [11], and green manure [12] have been reported to be effective.

Prefectures also recommend chemical analysis of the soil prior to planting. The Hokkaido fertilizer standard is determined by three levels of N fertility based on the analysis of available N (hot-water-extractable N) [13]. In the United States, the Pre-sidedress Soil Nitrate Test [14], which determines the need for top dressing by measuring the available N at a depth of 0–30 cm, has been widely used in corn production and has also been shown to predict the effectiveness of top dressing in cabbage [15]. Agricultural land is vast and varied, with uneven N composition. Meanwhile, a fertilizer plan is often uniform over the field because soil analysis is performed on a small number of representative samples before or during planting. Such a plan cannot account for uneven soil fertility within the field, resulting in excess or deficient soil fertilizer composition, variability in growth, and low yields. Precise estimation of plant N status is critical for optimizing N fertilization management. However, non-destructive and accurate N diagnostic methods for vegetables are scarce. Therefore, rapid and direct measurements in the field, rather than in the laboratory, are required [16].

There are two approaches to achieving more precise N control in the field—crop diagnosis and soil diagnosis. The first approach, crop diagnosis technology, is based mainly on image information. Unmanned aerial vehicles (UAVs), commonly known as drones, have recently emerged as a new tool and are becoming increasingly popular. In South Korea, RGB [17] and multispectral [18] camera images have been used to estimate the yield of Chinese cabbage. In Japan, some studies have estimated cabbage plant coverage using a field measurement system that acquires images of crop growth and satellite pictures [19], while others have estimated the projected area of cabbage plants by photographing the field with an RGB camera mounted on a drone [20]. Drone image analysis, which estimates the diameter and weight of cabbage heads, has been widely used [21].

Using growth data, simulation models are being developed for growth and yield prediction and cultivation support for open-field vegetables, such as cabbage in Europe [22] and Chinese cabbage in South Korea [23,24]. In Japan, Okada and Sasaki [25] reported that a cabbage growth model that estimated dry matter production, partitioning rate to the head, and shoot weight gain per dry weight increase as a function of temperature using solar radiation, average daily temperature, and planting density as input parameters could simulate dry matter production with a goodness of fit with less than a 5% relative error. Sugahara et al. [26] expanded this simulation by creating a program calculating the projected leaf area for each cabbage plant from images taken. These data were input along with weather data to estimate the expected harvest date and yield. Thus, crop growth diagnostic technology based on images of vegetables has started gaining attention for use in work coordination and shipment adjustment.

Methods to utilize crop growth data for soil management have been reported for rice, based on the normalized difference vegetation index, estimated by a small optical sensor (GreenseekerTMcanopy sensor) [27], and for Chinese cabbage, using leaf area index, dry weight, and N content estimated based on RGB camera images [28]. In Japan, variable-rate fertilizer application allows farmers to deliver different rates of fertilizer to each part of the field in real time. The method has been introduced into fertilizer top dressing for wheat and is now being incorporated into basal fertilizer applications for sugar beet and potato crops.

As an alternative approach, directly measuring soil inorganic N content in the field and diagnosing the timing and amount of fertilizer application, still needs to be improved. Field-installed $NO_3/NH_4$ ion-selective electrode sensor networks require multiple points to enable accurate measurements, and their efficacy should be carefully considered through cost–benefit analysis [29]. Rogovska et al. [30] reported the potential of diamond-attenuated

total internal reflectance Fourier-transform infrared spectroscopy as a soil nitrate sensor. Its accuracy was sufficient to divide the amount of fertilizer applied into two main levels but not enough to divide it into five levels.

Therefore, this study aimed to determine the methods and timing of crop and soil diagnosis in the early growth stage of cabbage to optimize the amount of N top dressing. We evaluated chlorophyll meter value, height, and projected leaf area as crop indicators, using water-content reflectometers (WCR) and a nitrate sensor as soil indicators. In addition, we discuss the potential for reducing within-field soil variability, yield uniformity, and fertilizer application through variable-rate N top dressing based on diagnostics.

## 2. Materials and Methods

### 2.1. Trial and Sampling Details

Three trials were conducted in this study, with Trials 1 and 2 aimed at data acquisition and diagnostic method generation, while Trial 3 was aimed at method verification. The experiments were conducted in a field (soil type: light-colored Andosol) at the National Agriculture and Food Research Organization (Tsukuba, Japan) for one year. The field was located at a latitude of $36°01'29''$ N and longitude of $140°06'17''$ E, 22 m above sea level. The size of the experimental site was $10.8 \times 28$ m for all trials. Sorghum was planted from May to July 2021, and the aboveground parts were removed. The electrical conductivity (EC) and pH of the crop soil (0.5 to 15 cm deep) before fertilization were 0.2 dS $m^{-1}$ and 5.5, respectively. The apparent specific gravity of the crop soil at transplantation, determined using the gravimetric method, was 0.74 at 5 to 10 cm deep and 0.80 at 12.5 to 17.5 cm deep. The readily available water content of the crop soil corresponding to pF l.5 to 3.0, determined using the sand column, pressure plate, and centrifugal methods, was 0.11 $m^3$ $m^{-3}$.

The cabbage plant (*Brassica oleracea* L. var. *capitata*) variety "Okina SP" (Takii Co., Ltd., Kyoto, Japan) was tested. Seedlings were transplanted 31–35 days after sowing in 128-hole plug trays filled with potting soil (Vegetable S type, Yanmar Holdings Co., Ltd., Tokyo, Japan), with 40 cm between plants and 60 cm between rows (planting density, 4.17 plants $m^{-2}$).

Weather measurements were taken using temperature and humidity sensors (HMP60, Vaisala Inc., Vantaa, Finland) inside natural ventilation tubes attached to the posts, a pyranometer (MS-602, Eko Instruments Co., Ltd., Tokyo, Japan), and a tipping bucket rain gauge (TR-525M, Texas Electronics Inc., Dallas, TX, USA). The sum of the average daily air temperature from the transplantation date to the day before harvest was considered the cumulative temperature. The soil temperature at a depth of 15 cm was measured using a temperature logger (RTR-502, TR-5530, T&D Corporation, Matsumoto, Japan). A WCR HydraProbe (HP) (5.7 cm long rod, Stevens Water Monitoring Systems Inc., Portland, OR, USA) was buried 15 cm deep within the fertilized plot in Trial 1 and logged soil volumetric water content (VWC), bulk EC, real and imaginary parts of the complex dielectric permittivity, as well as soil temperature were measured every 5 min.

At the end of Trial 3, three soil samples were collected from the crop soil of each plot, mixed, and air-dried. The soil and water were mixed in a ratio of 1:5, and EC was measured using an EC meter (LAQUAtwin, EC-33B, Horiba Advanced Techno Co., Ltd., Kyoto, Japan). Nitrate and ammonium concentrations were measured using ion chromatography (DX-320J, Thermo Fisher Scientific Inc., Tokyo, Japan). The soil was extracted with 10 times the volume of potassium chloride solution. Nitrate and ammonium concentrations were measured using an NC analyzer (NC-900, Sumitomo Chemical Co., Ltd., Tokyo, Japan).

The apparent utilization rate of N fertilizer in the fertilized plot was calculated using the subtraction method (Equation (1)):

$$
\begin{aligned}
&Apparent\ utilization\ rate\ of\ fertilizer\ N \\
&= (N\ absorption\ in\ fertilized\ plot - N\ absorption\ in\ non-fertilized\ plot) \\
&\div (Amount\ of\ fertilizer\ N) \times 100.
\end{aligned} \tag{1}
$$

*2.2. Selection of Suitable Crop and Soil Measurements for Diagnosis (Trials 1 and 2)*

Trials 1 and 2 were conducted 30 m apart. A 6 × 3 m non-fertilized plot was established within each trial, and the other area was fertilized. Crop residue compost of 2000 g m$^{-2}$ and magnesium carbonate fertilizer of 111 g m$^{-2}$ were applied to both trials on 6 August 2021. The fertilized plot received slow-release fertilizer (CDU-S555, N-P$_2$O$_5$-K$_2$O = 15-15-15, Jcam Agri. Co., Ltd., Tokyo, Japan), at a rate equivalent to 30 g N m$^{-2}$, 30 g P$_2$O$_5$ m$^{-2}$, and 30 g K$_2$O m$^{-2}$, 1 to 3 days before transplantation. Seeds were sown in Trials 1 and 2 on 16 July and 2 August 2021, respectively. The seedlings were transplanted on 19 August and 6 September 2021 in Trials 1 and 2, respectively.

After transplantation, soil measurements were performed twice a week. In advance, cylinders of vinyl chloride pipe (length, 12 cm; diameter, 13 mm) were buried vertically in the center between the plants (26 in fertilized plots and 7 in non-fertilized plots). Nitrate concentration was measured by inserting a soil nitrate meter (PRN-41, Fujiwara Scientific Co., Ltd., Tokyo, Japan) into the soil at a depth of 10 cm to the bottom of the cylinder for approximately 2 min. The soil nitrate meter was calibrated at two points, 100 mg NO$_3$-N L$^{-1}$ and 1000 mg NO$_3$-N L$^{-1}$, after the lacquer electrode membrane was infiltrated with the standard solution for 1 h. The purpose was to stabilize the variation of output values to ±2.5 mV (±15% error for nitrate concentration). The accuracy of the soil nitrate meter is reported to be good at a 30–50% water content in crop soil [16], which corresponds to a VWC of 0.39–0.65 m$^3$ m$^{-3}$ in the soil of this experiment. When the VWC was lower than 0.4 m$^3$ m$^{-3}$, the readings of the soil nitrate meter did not stabilize, so the measurements for that day were not utilized.

Two different WCR sensors, CS655 (CS) (12 cm long rod, Campbell Scientific Inc., Logan, UT, USA) and HP, were inserted vertically into the soil surface at the row center between the plants and measured for approximately 2 min. The calibration of the VWC of the sensor was based on an equation developed in a laboratory. The air-dried crop soil was weighed, and 900 g was placed into cylindrical plastic containers (diameter, 12.4 cm; height, 17.1 cm). Deionized water was added to reach 9 VWC ranging from 0.15 to 0.70 m$^3$ m$^{-3}$. To prepare samples with a low VWC, air-dried soil was placed in a plastic bag, deionized water was added, and the bag was shaken well before filling the cylindrical container. The solution was poured into the container to match the cylinder height to achieve the same density. The cubic regression equation was obtained using the least-squares method with the optimization analysis tool (solver) in Microsoft®Excel®for Microsoft 365 MSO, ver. 2208, from the real dielectric permittivity of the sensor output. The VWC was obtained by weight for CS and HP, which were buried entirely or inserted with a metal needle as the sensor installation style in the field (Table S1), respectively. Each coefficient of determination of the regression equation was higher than 0.97, indicating a good fit to the measured values.

The estimated soil solution EC was calculated using Equation (2) [31]:

$$\text{Soil solution EC} = \frac{\text{Real dielectric content of pure water} \times \text{Bulk EC measured with sensor}}{\text{Real dielectric permittivity of the soil} - \text{Real dielectric permittivity of dry soil}}. \qquad (2)$$

The real dielectric content of pure water was calculated from the soil temperature [32]. For each installation method, the real dielectric permittivity of the dry soil was entered as the actual measured value of the air-dried soil. The values were 2.83 for the CS-insert sensor, 3.69 for the HP-insert sensor, and 3.76 for the HP-buried sensor.

The soil solution was collected using the falling pressure method with porous cups (Mizutol for field use, DIK-8392, Daiki Rika Kogyo Co., Ltd., Kounosu, Japan) placed in the row center between plants (*n* = 4 for fertilized plots and *n* = 3 for non-fertilized plots). The collected soil solutions were frozen and stored, and the EC and ion concentrations were measured as described above.

The height, leaf color, and projected leaf area of individual cabbages (*n* = 25 for fertilized plots and *n* = 6–7 for non-fertilized plots) were surveyed twice weekly from transplantation to the beginning of head formation. Height was measured manually using

a measuring tape. Leaf color (SPAD value) (SPAD-502 Plus, Konica Minolta Japan Inc., Sakai, Japan) was measured at three locations on the second or third leaf from the top. A camera (GoPro HERO9 Black, Tajima Motor Corp., Tokyo, Japan) attached to a cart (Rakutaro RA-100N, Harax Co., Shibukawa, Japan) was used to photograph the cabbage from a height of approximately 51 cm. The projected leaf area was determined by image processing using the method described by Oishi et al. [33] (Figure S1).

The cabbages were harvested on 4 November in Trial 1 and 29 November in Trial 2. The harvested cabbage was measured in terms of head diameter, height, and fresh weight (FW). The dry weight was measured after drying (70 °C for 96 h). Dry matter was finely ground, and total N concentrations were measured using an NC analyzer.

### 2.3. Verification of the Effect of Variable-Rate N Top Dressing Based on Soil Diagnosis (Trial 3)

After the completion of Trial 2, Trial 3 was conducted at the same site. Treatment plots are shown in Figure 1.

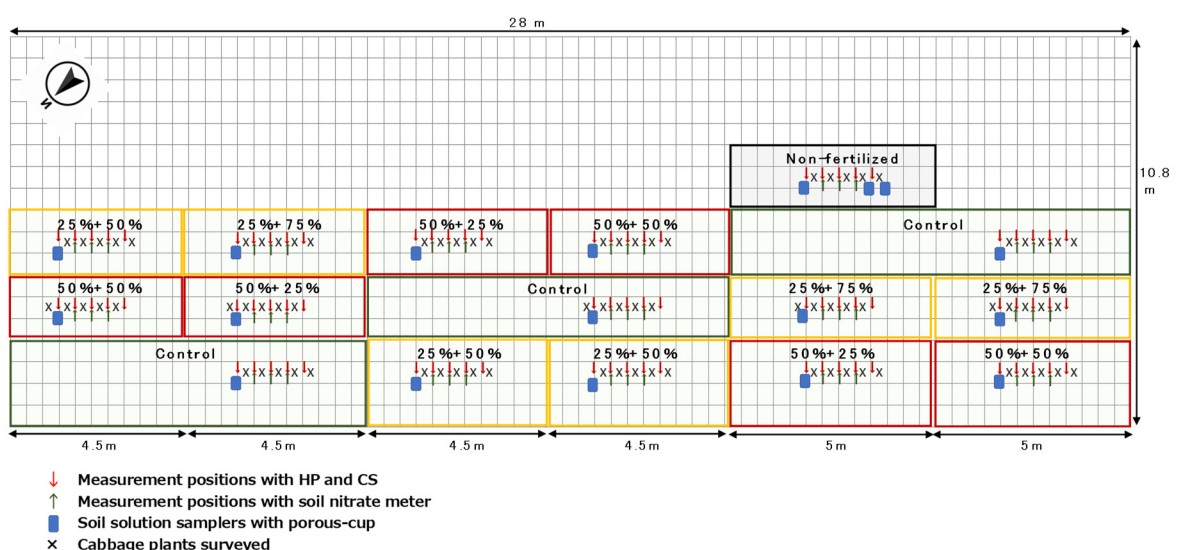

**Figure 1.** Layout of treatments and sensing positions (Trial 3). HP, HydraProbe; CS, CS655.

Crop residue compost was applied at 2000 g m$^{-2}$ on 21 February 2022, and magnesium carbonate fertilizer of 111 g m$^{-2}$ was applied on 22 February 2022. Basal fertilizer was applied to each plot 2 days before transplantation. A control plot was treated only with basal fertilizer using slow-release fertilizer (CDU-S555) at 30 g N m$^{-2}$, 30 g P$_2$O$_5$ m$^{-2}$, and 30 g K$_2$O m$^{-2}$. To provide variable-rate N top dressing plots, the amount of basal fertilizer was set at two levels, 25% or 50% of the N in the control plot. The N fertilizer was applied as granular urea (N 46, Mitsui Chemicals Inc., Tokyo, Japan) in amounts equivalent to 7.5 g N m$^{-2}$ at 25% and 15 g N m$^{-2}$ at 50%. Phosphoric acid and potassium were applied as a complex fertilizer (Mitsui PK, P$_2$O$_5$-K$_2$O=15-15, Kantodenko Co., Ltd., Takasaki, Japan) in amounts equivalent to 30 g P$_2$O$_5$ m$^{-2}$ and 30 g K$_2$O m$^{-2}$ for all plots except the non-fertilized plot. A randomized block design was used with three replicates. Each plot measured 8.1–12.0 m$^2$ (3 or 4 rows × 4.5 or 5 m long). As a reference, a non-fertilized plot was established at the same location as in Trial 2. Cabbage was sown on 28 February 2022, and transplanted on 31 March 2022.

As in Trials 1 and 2, a multipoint survey was conducted. The process included soil solution sampling and component analysis via ion chromatography (one point per plot), cabbage projected leaf area measurement (five individuals per plot), nitrate concentration measurement using a soil nitrate meter (three locations per plot), as well as VWC and soil solution EC measurement using HP and CS (five locations per plot).

The amount of N-top-dressing fertilizer was determined for each plot using the estimated soil solution EC of CS 14 days after transplantation (DAT). The standard was

established from the soil solution EC value of 1.0 dS m$^{-1}$ in the fertilized plot at the time of top dressing application in Trial 1 and 2 and that of 0.22 dS m$^{-1}$ in the non-fertilized plot at the same time in Trial 3. The standard was set as 100% for 0.22 to 0.42 dS m$^{-1}$, 75% for 0.42 to 0.62 dS m$^{-1}$, 50% for 0.62 to 0.82 dS m$^{-1}$, and 25% for 0.82 to 1.02 dS m$^{-1}$. Therefore, nitrogen was applied in sets of 25 + 50%, 25 + 75%, 50 + 25%, 50 + 50%, and a control. Each treatment had three replicates. In the variable-rate N-top-dressing plots, urea was scattered on the soil surface just before the cabbage plants covered the rows, which took place on 19 April (19 DAT), followed by intertillage. The cabbages were harvested and assessed on 4 July.

### 2.4. Statistical Analyses

All statistical analyses were performed using EZR (Saitama Medical Center, Jichi Medical University, Saitama, Japan, version 1.55) [34], which is based on R (The R Foundation for Statistical Computing, Vienna, Austria, version 4.1.2). More precisely, it is a modified version of the R commander (version 2.7-1) designed to add statistical functions frequently used in biostatistics.

## 3. Results

### 3.1. Climate and Soil Environment

The cabbage plants received a cumulative temperature of 1574 °C and 1390 °C, with a precipitation of 374 mm and 330 mm in Trial 1 and Trial 2, respectively (Figure 2). The average daily temperature for the 10 days after transplantation (19 August) in Trial 1 remained above 25 °C, with little precipitation. However, intermittent rainfall occurred before and after transplantation (6 September) in Trial 2, and the average temperatures remained low, ranging from 19 to 24 °C. Two significant rainfall events occurred in September and early October, but there were also many rainless days, and temperatures remained above normal. From mid-October onward, the temperatures were near normal, and there were many sunny days. The VWC of soil at a depth of 15 cm, measured by the HP buried in the fertilized plot of Trial 1 ranged from 0.42 to 0.60 m$^3$ m$^{-3}$, indicating sufficient water. The EC in the fertilized plot, measured by an HP, ranged from 0.31 to 1.80 dS m$^{-1}$. Conversely, the measured soil solution EC in the fertilized plot, which was collected in 15-cm-deep porous cups, ranged from 0.32 to 2.50 dS m$^{-1}$, being higher than sensor estimates in the first half of the growing period. Both the estimated and measured soil solution EC in the fertilized plot peaked from the end of August to the beginning of September and then decreased, remaining below 0.6 dS m$^{-1}$ from late September onward. The measured soil EC in the non-fertilized plots ranged from 0.11 to 0.30 dS m$^{-1}$.

The cabbage plants received a cumulative temperature of 1728 °C and a precipitation of 382 mm in Trial 3 (Figure S2). The average temperatures in April and June were considerably higher than normal. Precipitation was higher than normal in April and May but much lower than normal in June, with no rainfall exceeding 10 mm after 12 June.

Among ion concentrations in the collected soil solutions of the fertilized plots in Trials 1 and 2, the contribution of NO$_3$-N was most significant for anions up to DAT 40 (Figure 3). Thereafter, all anions were present at low concentrations, and the EC was also low. For cations, ion contributions were in the order of Ca > Mg, K > NH$_4$-N, and Na (data not shown). NO$_3$-N concentration in the fertilized plot was 223 mg L$^{-1}$ in Trial 1 and 166 mg L$^{-1}$ in Trial 2 at the maximum value. However, all ion concentrations were diluted in the non-fertilized plots, especially in Trial 1, where NO$_3$-N and SO$_4$-S contributed similar amounts in the early stages, and SO$_4$-S was the major anion after mid-September (data not shown). NO$_3$-N concentrations in the non-fertilized plots were low, with maximum values of 7 mg L$^{-1}$ in Trial 1 and 24 mg L$^{-1}$ in Trial 2.

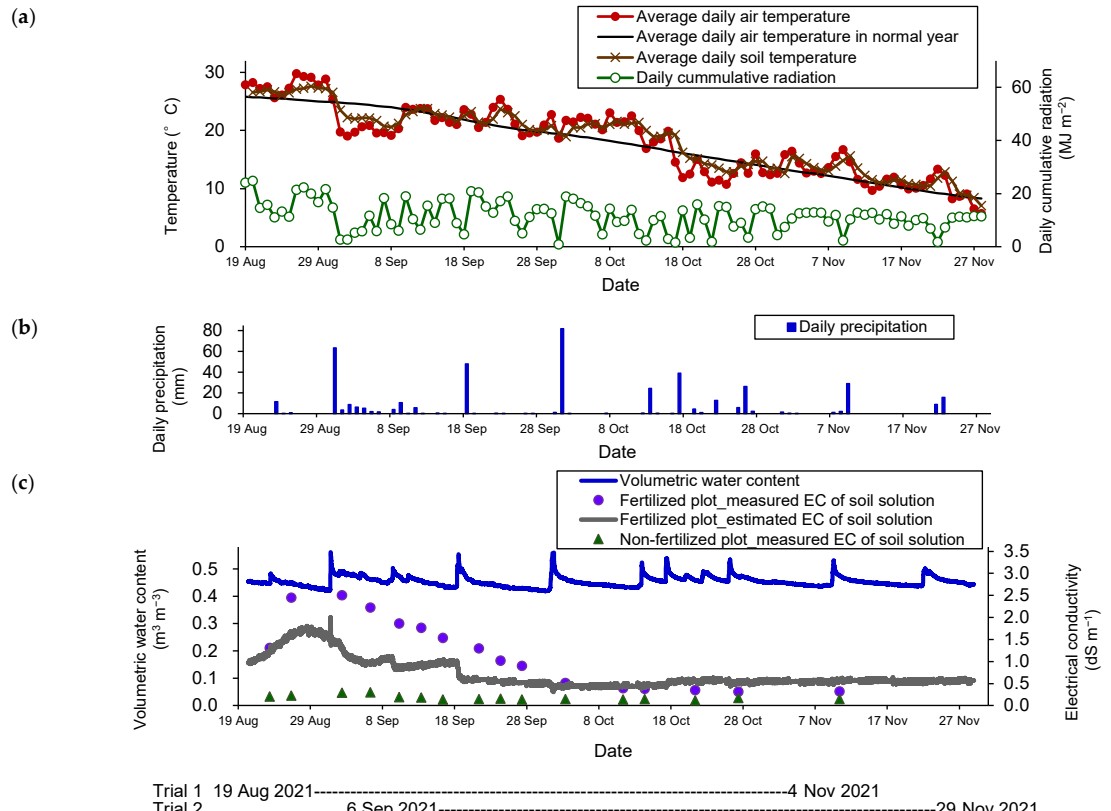

**Figure 2.** Climatic and soil environment during the growing seasons of Trials 1 and 2. (**a**) Air temperature, soil temperature, and cumulative radiation; (**b**) precipitation; (**c**) volumetric water content and measured or estimated soil solution electrical conductivity (EC). Estimated soil solution EC was calculated by an HP buried at a depth of 15 cm. The measured EC values represent the mean of four (fertilized plot) or three (non-fertilized plot) replicates.

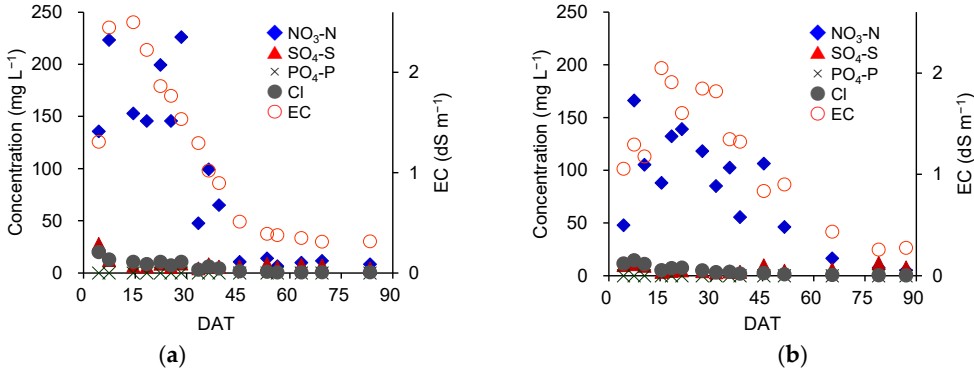

**Figure 3.** Electrical conductivity (EC) and anion concentrations of soil solutions in the fertilized plot during the growing seasons of Trial 1 (**a**) and Trial 2 (**b**). Each plot represents the mean of four replicates.

### 3.2. Cabbage Yield and N Absorption

The cabbage characteristics at harvest were significantly different for all survey indicators between the fertilized and non-fertilized plots in Trials 1 and 2 (Table 1). Thus, the FW of the head in the non-fertilized plot was 25% lower than that in the fertilized plot in Trial 1 and 51% lower than that in the fertilized plot in Trial 2. Although the difference in the FW of the total shoots was less than that in the FW of the head, the FW of the cabbage head in the fertilized plot greatly exceeded that in the non-fertilized plot in both trials. N

content was 2.9 to 3.0% in the head, 3.5% in the outer leaves, and 3.3 to 3.9% in the stem of plants in the fertilized plot, all being more than 0.5% higher than those in the non-fertilized plot. The total shoot N uptake calculated from the N content and dry weight of each plot was 4.7 g plant$^{-1}$ (Trial 1) and 4.9 g plant$^{-1}$ (Trial 2) in the fertilized plot and 1.7 g plant$^{-1}$ (Trial 1) and 2.4 g plant$^{-1}$ (Trial 2) in the non-fertilized plot. The apparent utilization rate of fertilizer N calculated via Equation (1) in the fertilized plot was 41% in Trial 1 and 35% in Trial 2.

**Table 1.** Effects of fertilization on the yield of cabbage and nitrogen content (Trials 1 and 2). Values are the median values obtained from 25 replicates for fertilized plots and 6–7 replicates for non-fertilized plots (quartile range). *p*-values from the Mann–Whitney U test are shown.

| Trial | Treatment | Fresh Weight (g plant$^{-1}$) | | | | Nitrogen Content (% on a Dry Weight Basis) | | | | | | Plant Nitrogen (g plant$^{-1}$) | |
|---|---|---|---|---|---|---|---|---|---|---|---|---|---|
| | | Head | | Total Shoots | | Head | | Outer Leaves | | Stem | | Total Shoots | |
| Trial 1 | Fertilized | 1374 | (1196–1492) | 2176 | (1888–2270) | 3.0 | (2.7–3.1) | 3.5 | (3.4–3.6) | 3.9 | (3.6–4.3) | 4.7 | (4.2–5.0) |
| | Non-fertilized | 343 | (278–421) | 817 | (756–983) | 2.5 | (2.3–2.6) | 2.1 | (2.0–2.1) | 2.4 | (2.1–2.4) | 1.7 | (1.6–2.0) |
| | *p*-values | <0.001 | | <0.001 | | <0.001 | | <0.001 | | <0.001 | | <0.001 | |
| Trial 2 | Fertilized | 1277 | (1162–1420) | 2146 | (1914–2284) | 2.9 | (2.8–3.2) | 3.5 | (3.4–3.5) | 3.3 | (3.2–3.6) | 4.9 | (4.5–5.2) |
| | Non-fertilized | 651 | (564–823) | 1318 | (1096–1511) | 2.2 | (2.0–2.3) | 2.0 | (1.9–2.4) | 2.0 | (1.9–2.1) | 2.4 | (2.1–2.7) |
| | *p*-values | <0.001 | | <0.001 | | <0.001 | | <0.001 | | <0.001 | | <0.001 | |

### 3.3. Correlation of Crop Growth Indicators with Yield at Early Stages (Trials 1 and 2)

Spearman's rank correlation coefficients were determined between some growth indicators for cabbage during the early stages of growth, and the FW of the heads and total shoots at harvest (Table 2). In Trials 1 and 2, SPAD values at the seedling stage showed a weak positive correlation with head FW and total shoot FW but no positive correlation at the rosette stage. Height at the seedling stage showed no positive correlation with head FW and total shoot FW, while height at the rosette stage showed a weak positive correlation in both trials. The projected leaf area and its rate of increase had large correlation coefficients between 0.571 and 0.672 with head and total shoot FW at both growth stages in both trials. Correlation coefficients between the projected leaf area or its rate of increase and head FW were greater than those between the projected leaf area and the total shoot FW.

### 3.4. Comparison of Measurement Sensors for Variable-Rate N Fertilization (Trials 1 and 2)

Multipoint measurements of soil nitrate concentration and soil solution EC assessed via the HP and CS continued up to 36 DAT (Trial 1) and 30 DAT (Trial 2). These values peaked by the third week after transplantation in both trials (Figure 4). The soil nitrate concentrations varied greatly between sites. For example, at 15 DAT in Trial 1, when the maximum values were observed at many sites, these ranged from 177 to 2423 mg L$^{-1}$ in the fertilized plots and from 23 to 355 mg L$^{-1}$ in the non-fertilized plots. On the same day, EC values measured by the HP ranged from 0.32 to 0.54 dS m$^{-1}$ in the fertilized plot and from 0.27 to 0.30 dS m$^{-1}$ in the non-fertilized plot. In comparison, the EC measured by the CS ranged from 0.25 to 0.59 dS m$^{-1}$ in the fertilized plot and from 0.11 to 0.14 dS m$^{-1}$ in the non-fertilized plot, showing less variation than the nitrate concentration. The EC measured by HP dropped sharply after the maximum value, while the EC measured by the CS decreased relatively slowly in Trials 1 and 2.

**Table 2.** Spearman's rank correlation coefficients of crop and soil indicators during early growth stages with the head and shoot fresh weight at harvest (Trials 1 and 2). Values are the medians of 31 replicates for Trial 1 and 32 replicates for Trial 2. FW, fresh weight.

| Growth Stage | Indicator | Trial 1 | | Trial 2 | |
|---|---|---|---|---|---|
| | | Head FW | Total Shoot FW | Head FW | Total Shoot FW |
| Seedling [a] | SPAD | 0.402 * | 0.416 * | 0.510 ** | 0.440 * |
| | Height | −0.014 | 0.018 | 0.246 | 0.094 |
| | Increase in projected leaf area | 0.658 *** | 0.591 *** | 0.595 *** | 0.577 *** |
| | Projected leaf area | 0.672 *** | 0.592 *** | 0.599 *** | 0.571 *** |
| | Soil $NO_3$-N | 0.41 * | 0.431 * | 0.574 *** | 0.485 ** |
| | Soil EC with HP sensor | 0.396 * | 0.417 * | 0.456 ** | 0.342 |
| | Soil EC with CS sensor | 0.519 ** | 0.554 ** | 0.364 * | 0.277 |
| Rosette [b] | SPAD | −0.386 | 0.100 | −0.002 | −0.109 |
| | Height | 0.478 ** | 0.467 ** | 0.476 ** | 0.351 * |
| | Increase in projected leaf area | 0.621 *** | 0.593 *** | 0.629 *** | 0.619 *** |
| | Projected leaf area | 0.644 *** | 0.587 *** | 0.649 *** | 0.647 *** |
| | Soil $NO_3$-N | 0.458 *** | 0.429 * | 0.255 | 0.140 |
| | Soil EC with HP sensor | 0.470 * | 0.458 * | 0.402 * | 0.284 |
| | Soil EC with CS sensor | 0.510 ** | 0.565 ** | 0.424 * | 0.323 |

Asterisks indicate significant correlations: * $p < 0.05$, ** $p < 0.01$, and *** $p < 0.001$. [a], 7-15 DAT. [b], 15-21 DAT.

Spearman's rank correlation coefficient was used to test the relationship of soil measurements with head and total shoot FW at harvest (Table 2). Positive correlations were observed between the soil nitrate concentration or EC measured by HP and the head and total shoot FW at both growth stages in Trial 1. However, in Trial 2, there was either no correlation or smaller values were observed at the rosette stage compared with those at the seedling stage. However, the correlation coefficients between EC measured by the CS and head or total shoot FW were greater at the rosette stage than at the seedling stage in both trials, and the correlation with head FW was significant in both cases. Therefore, in Trial 3, the EC measured by the CS was employed for soil diagnosis for variable-rate N top dressing.

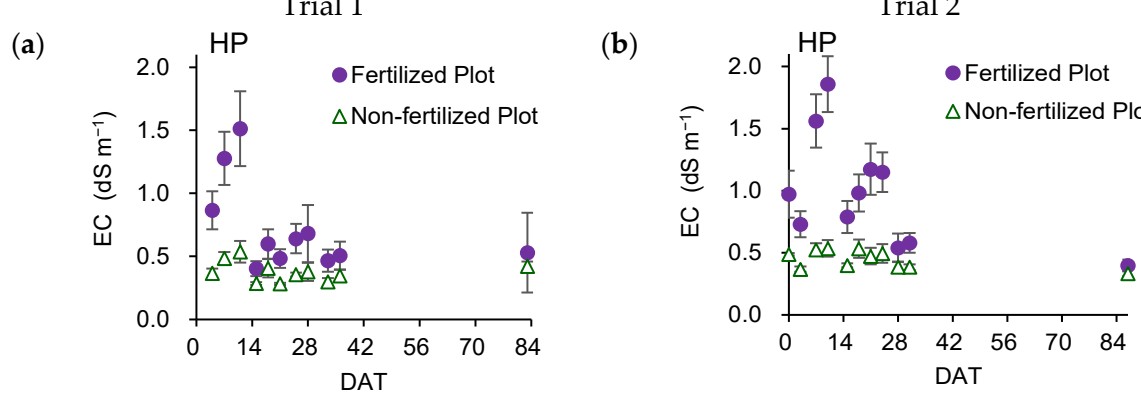

**Figure 4.** *Cont.*

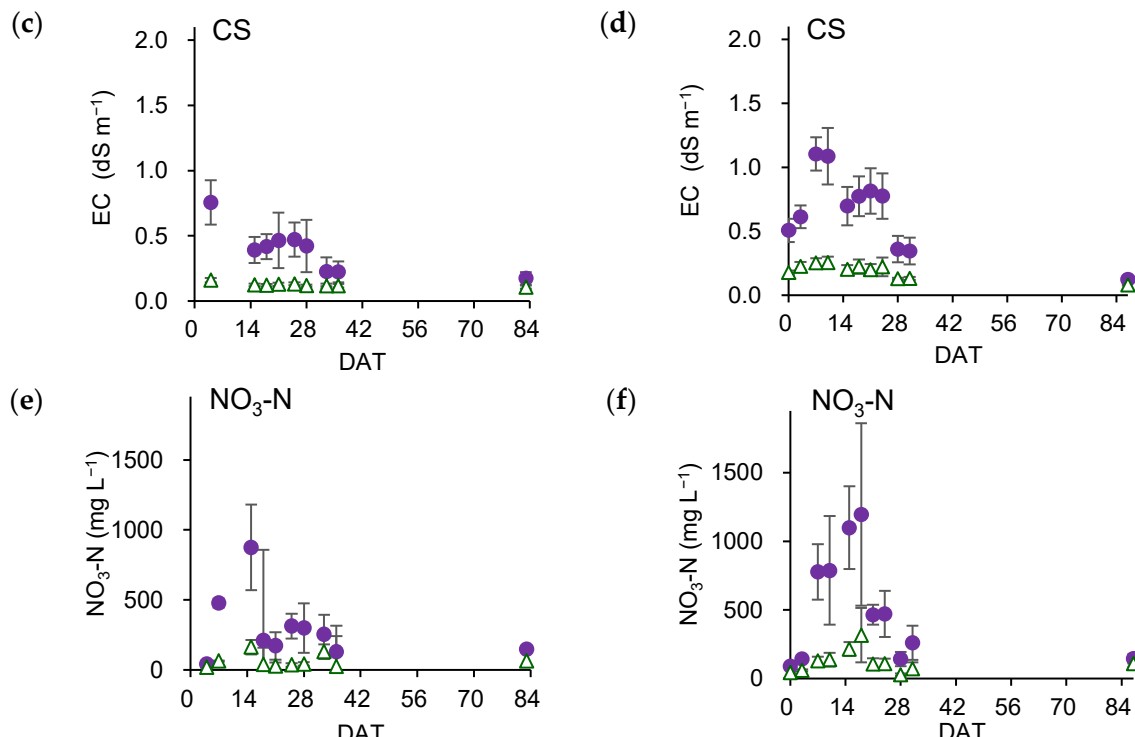

**Figure 4.** Electrical conductivity (EC) measured by the HP (**a,b**), EC measured by the CS (**c,d**) and nitrate concentration measured by the soil nitrate meter (**e,f**) at multiple points (Trial 1 and 2). Values represent the mean obtained from 25 (fertilized plot) or 6–7 (non-fertilized plot) replicates. Bars represent standard deviation (SD). EC measured by the CS on 7 DAT in Trial 1 is missing.

### 3.5. Verification of the Effect of Variable-Rate N Top Dressing Based on Soil Diagnosis (Trial 3)

The EC measured using the CS gradually increased after transplantation (Figure 5a). Moreover, sensing values of the EC at 14 DAT were used to determine the fertilizer amount. The EC values were divided into four ranges between the EC in the non-fertilized plot (0.22 dS m$^{-1}$) and the maximum EC value in the fertilized plot in Trial 2 (1.02 dS m$^{-1}$, Figure 4e): 0.22 to 0.42 (dS m$^{-1}$):30 (g N m$^{-2}$), 0.42 to 0.62:22.5, 0.62 to 0.82:15, and 0.82 to 1.02:7.5. A variable fertilizer top dressing was applied per plot at 19 DAT (19 April), according to the average EC (*n* = 5) of each plot (Figure 6).

After top dressing fertilization at 21–28 DAT, the soil solution EC measured by the CS sensor showed an increasing trend, and the degree of increase was greater in the 50 + 50% plot than in the 50 + 25% plot as well as in the 25 + 75% plot than in the 25 + 50% plot. This is presumably due to elution of the additional urea fertilizer. After 21 days, there were no significant differences in the EC among treatments.

The nitrate concentrations in the collected soil solution were higher than 350 mg L$^{-1}$ in the control at the mid-growth stage (Figure 5b). However, there was a large variation among the three replicates and no significant differences among the treatments for any of the sampling dates. In this study, soil diagnosis was conducted at 14 DAT in all treatments after artificially changing the amount of basal fertilizer N. Figure 6 shows that the increase in the projected leaf area of seven plots was lower than the average value (13.1 cm$^2$ day$^{-1}$). Four of those plots, including the non-fertilized plot had low EC. Therefore, by narrowing down the soil measurement points, where the increase in projected leaf area was lower than the average value, it was possible to detect three plots with low EC among the plots with 25% basal fertilizer (Figure 6). The variable-rate of N top dressing based on the reference criterion of EC values set at 14 DAT reduced the subsequent variation in EC between plots in the soil (Figure 5). There were no significant differences in the final cabbage head FW or total shoot FW after treatment (Table 3). The N content was lower in the 50 + 50% plot

than in the control only in the outer leaves, but there was no difference in N absorption of the total shoots due to treatment. Nitrogen uptake ranged from 24 to 28 g m$^{-2}$ in terms of area. Apparent fertilizer N utilization, as determined in Trials 1 and 2, was lowest in the 25 + 75% plot at 39%, followed by the 50 + 50% plot at 46%, 50 + 25% plot at 61%, and 25 + 50% plot at 62%. In contrast, the control plot with slow-release fertilizer had 50% apparent N fertilizer utilization. Nitrate and ammonium extracted via potassium chloride remaining in the crop soil at the end of cultivation were low in plots with variable-rates of N, as in the control plot (Table 4).

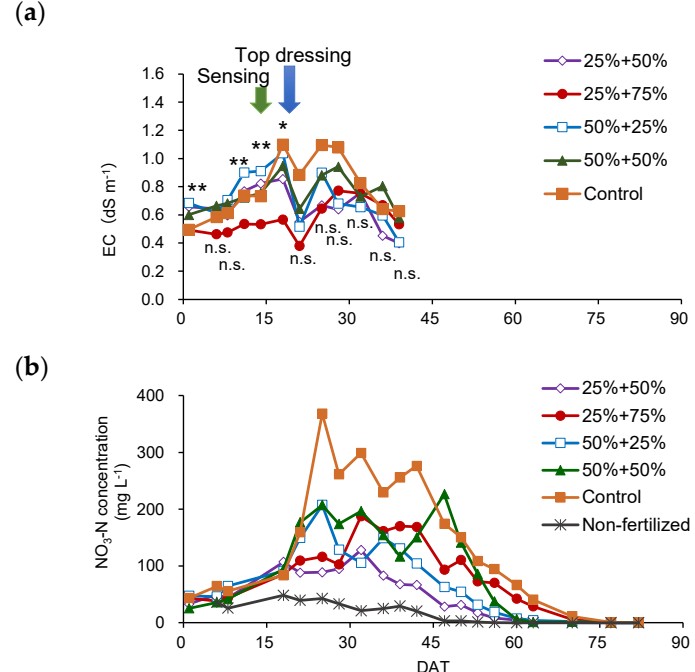

**Figure 5.** EC measured by the CS (**a**) and nitrate concentration in soil solution (**b**) (Trial 3). Values represent the mean values obtained from three replicates. Asterisks indicate significant differences: * $p < 0.05$ and ** $p < 0.01$. n. s. indicate no significance ($p > 0.05$). Nitrate concentration: no significant difference for all days except in the non-fertilized plot.

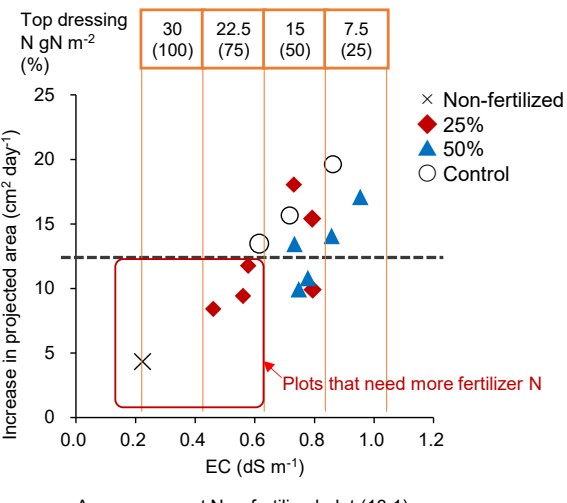

**Figure 6.** Relationship between EC measured by the CS and increase in the projected leaf area (Trial 3). Values represent the mean of values obtained from five replicates. Increase in the projected leaf area was between 14 and 18 DAT; EC was measured on 14 DAT. Dotted line represents the average except for the non-fertilized plot.

**Table 3.** Effects of variable fertilization on the yield of cabbage and nitrogen content (Trial 3). Medians followed by the same letter are not significantly different according to the Kruskal–Wallis test. n.s. indicates no significance at $p < 0.05$.

| Treatment Basal + Top Dressing | Fresh Weight [a] (g Plant⁻¹) | | | | Nitrogen Content [b] (% on a Dry Weight Basis) | | | | | | Plant Nitrogen [b] (g Plant⁻¹) | |
|---|---|---|---|---|---|---|---|---|---|---|---|---|
| | Head | | Total Shoot | | Head | | Outer Leaves | | | Stem | | Total Shoot | |
| 25 + 50% | 3018 | (2821–3219) | 4041 | (3865–4152) | 3.0 | (2.7–3.1) | 2.7 ab | (2.0–2.8) | 3.8 | (3.5–4.2) | 6.2 | (6.2–7.0) |
| 25 + 75% | 2749 | (2583–3010) | 3544 | (3437–3967) | 2.9 | (2.7–3.1) | 2.7 ab | (2.2–2.9) | 3.9 | (3.6–4.0) | 5.8 | (5.5–6.4) |
| 50 + 25% | 2903 | (2740–3036) | 3738 | (3617–4057) | 3.1 | (3.0–3.2) | 2.6 ab | (2.3–2.7) | 4.0 | (3.8–4.3) | 6.4 | (6.1–6.8) |
| 50 + 50% | 2959 | (2841–3138) | 3891 | (3724–4178) | 2.8 | (2.6–2.9) | 2.5 b | (2.2–2.6) | 4.0 | (3.7–4.1) | 6.6 | (5.8–6.8) |
| Control | 2917 | (2819–3130) | 3933 | (3680–4143) | 3.3 | (3.0–3.4) | 3.0 a | (2.2–3.0) | 3.9 | (3.6–4.2) | 6.6 | (6.2–7.2) |
| p-values | n.s. | | n.s. | | n.s. | | <0.01 | | | n.s. | | n.s. | |

[a] Values are the medians of 15 replicates (quartile range). [b] Values are the medians of nine replicates (quartile range). The same letters a and b indicate statistically insignificant differences ($p > 0.05$).

**Table 4.** Effects of variable fertilization on soil nitrate and ammonium concentrations obtained using potassium chloride extraction after the harvest (Trial 3). Values represent the median obtained from three replicates (standard deviation). $p$-values from the Kruskal–Wallis test are shown. n.s. indicates no significance at $p < 0.05$.

| Treatment Basal + Top Dressing | NO₃-N (g 100 g⁻¹ Dry Soil) | | NH₄-N (g 100 g⁻¹ Dry Soil) | |
|---|---|---|---|---|
| 25 + 50% | 0.82 | (0.19) | 1.23 | (0.08) |
| 25 + 75% | 1.30 | (1.23) | 1.10 | (0.09) |
| 50 + 25% | 0.59 | (0.03) | 1.17 | (0.10) |
| 50 + 50% | 0.46 | (0.04) | 1.30 | (0.19) |
| Control | 1.77 | (1.50) | 1.12 | (0.04) |
| p-value | n.s. | | n.s. | |

## 4. Discussion

### 4.1. Indicators and Timing of Crop Diagnosis

Previous reports have examined the use of SPAD as a diagnostic indicator for variable fertilizer application to head-forming vegetables. The Hokkaido prefectural government [35] reported a strong negative correlation between the SPAD values of cabbage and sugar content at harvest, suggesting that these could be used to determine the need for fertilizer application but only at the beginning of the head formation stage. Rongting et al. [36] found that the SPAD values of the unfolded leaves of Chinese cabbage showed no correlation with plant N absorption in the early growth stage (16–19 DAT) and a significantly strong positive correlation after the rosette stage (28–31 DAT). Herein, SPAD values differed in their correlation with final growth, depending on the timing of the measurement. Xiong et al. [37] provided evidence that the relationship between SPAD readings and N content per leaf area is profoundly affected by environmental factors and the leaf features of crop species. The results suggest that the time of SPAD measurements, environmental irradiance, and species must be considered to ensure precise monitoring of leaf N status with chlorophyll meters. In contrast, tissue NO₃-N concentrations are highly variable, and crop diagnosis based on critical N concentrations requires local research and field verification [38]. In this experiment, diagnosis during the early growth period was required to determine the variable N top dressing amount for cabbage. However, it should be noted that SPAD may produce erroneous results.

For the early stage assessed in this study, as with the SPAD values, plant height differed in terms of the presence or absence of a correlation with final growth, depending on the time of measurement. It was found that the measurement of cabbage height can be performed accurately from UAV images [39]. However, Kimchi (Chinese) cabbage height growth exhibits an S-shaped sigmoidal curve [23]. The initial two stages proceed at considerably different speeds, the first being slow and the second fast. The outer leaves develop horizontally in the early stage up to the beginning of cabbage head formation, so it is difficult to use the height for growth diagnosis during this period.

In contrast, the projected leaf area or the rate of increase in projected leaf area at both the seedling and rosette stages showed relatively strong correlations with the FW of the head and the total shoots at harvest. The results of this study are consistent with those of Tanaka et al. [20], who showed that the projected leaf area of individual cabbage plants at 15 and 20 DAT, calculated from images taken by a drone, had a high positive correlation with the FW of the head at harvest. In our study, the projected leaf area and its rate of increase had similar correlation coefficients with head FW or cabbage yield. However, good or poor seedling establishment influences the uniformity of subsequent growth [40]. This means that if there is considerable variation in seedling size, it is better to use the rate of increase in the projected leaf area rather than the projected leaf area itself in the early stages of growth.

These results suggest that the rate of increase in projected leaf area and the rosette stage (2–3 weeks after transplantation) are appropriate crop growth diagnostic indicators and timing for yield estimation, respectively, and can be useful for determining the variable N top dressing amount.

*4.2. Indicators and Timing of Soil Diagnosis*

There was either a weak or no correlation between the soil indicators and the final FW of the head. Soil diagnosis should be conducted to detect the excess or deficiency of N fertilizer because crops can take up a wide range of N concentrations in the soil. As Padilla et al. [41] noted in their review, the combination of crop and soil monitoring provides vegetable growers with tools to detect crop N deficiency and excess N supply.

The nitrate concentration of the soil solution collected in the crop soil with ceramic cup suction samplers can be used for N management of fertigated vegetables [41]. Padilla et al. mentioned that an important practical issue with ceramic cup suction samplers is the limited range of matric potentials ($0-50$ kPa) at which sampling is possible. This corresponds to a VWC of 0.70 to 0.43 $m^3\ m^{-3}$ for the soils in this study and is limited to after rainfall. The solution samples should be measured in the field simply using ion-selective electrodes or via laboratory analysis. Multipoint measurements of the soil solution are labor-intensive. Therefore, real-time and direct measurement of nitrate concentrations in the soil is of great significance for variable N fertilization. The soil nitrate meter tested in this study was an ion-selective electrode with a lacquer film as the matrix material. The developer [16] reported that the lower limit of soil moisture that can be measured with high accuracy by this sensor is 30% of water content by weight, which is 0.39 $m^3\ m^{-3}$ in terms of the VWC of the soil tested in this study. In fact, in late August (Trial 1), when the VWC was close to this value, the output value of the soil nitrate meter was unstable and could not be adopted. Low-moisture conditions of this magnitude often occur in field soils. In addition, the diameter of the lacrimal chip of the soil nitrate meter was small (5.8 mm), and the measurement was localized. Therefore, even at similar soil moisture levels, the measured values have a considerable variance owing to slight differences in the measurement location. Moreover, the soil nitrate meter cannot be inserted into hard soil, so a measurement hole must be dug each time. It also requires calibration with a standard solution just prior to measurement, making the method difficult to implement.

The contribution of nitrate to the EC of the soil solution was significant in the fertilized plot. Therefore, instead of the soil nitrate meter, EC values from WCR sensors are a potential alternative for determining the abundance of N content in the soil for cabbage plants. It would be best to measure the soil depth over time or frequently. However, measurements can be made on the root zone of cabbage plants before N top dressing so as to reduce the testing frequency. Most of the cabbage plant root system is distributed in the surface layer at 10 cm, although it varies with planting depth [42]. As CS has a longer rod (12 cm) than the HP (5.7 cm rod), it could cover the root zone at the early growth stage. Thus, the EC measured using CS was considered suitable for soil diagnosis.

Multipoint soil monitoring with WCR has drawbacks in terms of cost, calibration for each soil property [43], and data uncertainty due to sensor interreplicate variability [44].

In this study, we tested a method for measuring multiple points in the field using a single sensor. This requires carrying the sensor between these multiple points. Although no system is currently in practical use, some authors [45] are working on automating growth and soil measurements with a small robot. We plan to develop a system that will enable rapid diagnosis in the field and link it to agricultural machinery, for example, variable fertilizer application machines.

### 4.3. Effect of Variable N Top Dressing on Cabbage Yield and N Fertilizer Utilization

As described above, there was no significant difference between the fertilized treatments in head FW or apparent N uptake of the cabbage plants. The harvest survey was conducted at 95 DAT, and all plants except for those in the non-fertilized plot produced 2 kg or more of sample material, enough for processing operations. Variation in soil EC after top dressing fertilization remained small because fertilizer was applied according to the EC of the crop soil at 14 DAT, indicating that the soil N environment in the field became more uniform than previously. No difference in inorganic N concentration in the crop soil at the end of cultivation was observed with the variable fertilizer treatment. Although leaching was not investigated in this study, our data suggest that variation in the crop soil fertility did not increase. Therefore, variable N top dressing had no negative effect, especially when N fertilizer was reduced by 25% in the 25 + 50% and 50 + 25% plots. In this study, N fertilizer was reduced by 25% in half of the treatment plots, which was considered appropriate.

Apparent N fertilizer utilization in this study ranged from 35% to 50% in the slow-release fertilizer treatments in Trials 1–3 and from 39% to 62% in the variable fertilizer treatments with urea in Trial 3. This was comparable to the N utilization of slow-release fertilizers (36%) in a trial by Li et al. [7]. Everaarts and Booi [46] found that the maximum N uptake of cabbage studied at four locations in two cropping seasons in the Netherlands was approximately 40 g N m$^{-2}$, and the efficiency of N utilization for dry matter production decreased with increasing N application (0 to 42.5 g N m$^{-2}$). In our test, N utilization was slightly lower in the 25 + 75% plot, where the amount of applied top dressing fertilizer was high, suggesting that the fertilizer dose (22.5 kg), which was much higher than that of the base fertilizer (7.5 kg), was excessive. The standard value for the variable fertilizers should be optimized by accumulating more data in the future. The diagnostic method proposed in this study may not be reproducible under all conditions because of annual differences, especially with respect to weather conditions. This method can be applied when sufficient rainfall and soil temperature are available, and the effect of the base fertilizer on soil N content is normal.

### 5. Conclusions

To determine the variable N top dressing amount for cabbage plants, the projected leaf area and its increase rate were selected as crop diagnostic indicators. The projected leaf area and its growth rate at the rosette stage exhibited large correlation coefficients with head FW, whereas the soil nitrate concentration and solution EC peaked 3 weeks after transplantation. Thus, we determined the rosette stage 2 to 3 weeks after transplantation, as a suitable time for the survey. Two types of WCR sensors with different rod lengths and a soil nitrate sensor were compared as soil diagnostic indicators. A WCR sensor with a rod length of 12 cm, which is close to the crop soil depth, can estimate EC. The variance in inorganic N content in the field was reduced by using variable N fertilization based on the EC of the crop soil 2 weeks after transplantation. The proposed diagnostic method (estimation of EC using a WCR sensor with a rod comparable to the length of the root zone depth at the rosette stage) enables efficient local soil management while maintaining productivity. Taken together, the increase in projected leaf area in the early growth stage was used to narrow down the soil measurement points, and the WCR sensor was used to estimate the soil solution EC to determine the variable fertilizer amount. The application of field measurements combining crop and soil sensing requires further data accumulation

under different environmental conditions such as soil type and weather conditions. In addition, implementation would require the development of systems that enable rapid diagnosis in the field and linkage with agricultural machinery.

**Supplementary Materials:** The following supporting information can be downloaded at: https://www.mdpi.com/article/10.3390/horticulturae9040506/s1, Table S1: Relationship between real dielectric permittivity and volumetric water content of tested soil: coefficients of regression equation. Figure S1: Procedure for measuring the projected leaf area. (a) Input original image; (b) Extracted individual cabbage after binarization process. Figure S2: Climatic environment during the growing seasons of Trial 3. (a) Air temperature, soil temperature, and cumulative radiation; (b) Precipitation.

**Author Contributions:** Conceptualization, Y.N. and W.I.; formal analysis, Y.N.; investigation, Y.N., S.N., Y.M. and M.K.; resources, Y.N. and F.S.; writing—original draft preparation, Y.N.; writing—review and editing, S.N. and M.K.; supervision, W.I. and F.S.; project administration, Y.N. and Y.M.; funding acquisition, W.I. All authors have read and agreed to the published version of the manuscript.

**Funding:** This study was funded by the Public–Private R&D Investment Strategic Expansion Program (PRISM) of the Cabinet Office, Government of Japan.

**Data Availability Statement:** Restrictions apply to the availability of the data used, which were under license for the current study and hence are not publicly available. Data are, however, available from the corresponding author (Y.N.) upon reasonable request and with permission.

**Acknowledgments:** We express our sincere thanks to S. Ohno, M. Komada, and S. Kikuchi for their immense help in supporting the management of this project.

**Conflicts of Interest:** The authors have no conflicts of interest to declare.

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
