# Peer review of "Applicability of Variable-Rate Nitrogen Top Dressing Based on Measurement of the Within-Field Variability of Soil Nutrients for Cabbage Production"

_horticulturae, doi:10.3390/horticulturae9040506_

Round 1

Reviewer 1 Report

Dear authors,

I would like to congratulate you on your work. Please consider the following comments to improve your manuscript. Importantly, please is to highlight the originality and novelty of your contribution in the first line of the abstract, the last paragraph of the introduction, and in the conclusion section:

-          Line 12: the authors could include 1 additional line of background of their investigation. For example, as you explain in the introduction of your article (lines 52 to 75) the need for rapid direct measurements in the field rather than inconvenient laboratory procedures.

-          Lines 103 to 107: The last paragraph of the introduction where you are supposed to present the hypothesis of your investigation, or the objectives could be more specific. The information on which particular methods will you be evaluating in your article is missing.

-          Line 184: this equation should be numbered and cited in the text.

-          Could you check whether the first sentence in the last paragraph of the materials and methods is grammatically correct?  I am particularly talking about the coordinate clause in the sentence: All statistical analyses were performed using EZR (Saitama Medical Center, Jichi Medical University, Saitama, Japan, version 1.55) [34], which is for R (The R Foundation for Statistical Computing, Vienna, Austria, version 4.1.2).

-          Lines 247 and 267: How the cumulative temperature was calculated? Was it calculated over a period of 2 months? I guess this can be seen in Figure 2a but it would be helpful for the reader to be mentioned in the text.

-          Lines 302 to 304: This calculation should be better explained in the section of materials and methods. Also, this equation should be numbered and cited in the text.

-          Lines 305 to 307: Please, use a superscript to specify where the information of the footnote of the table is applicable. Otherwise, if it is general information for all the table I would suggest to include it in the title of the table.

-          Line 332 to 336: Please, align the table to the right side.

-          Line 352: Space is missing between the units “g Nm-2

-          Lines 357 to 362: DAT = Days After Planting, hence the beginning of the sentence: After 21 DAT is not correct because it is redundant. Furthermore, it is described in the text as “no significant differences in EC among treatments” but clearly there are some error bars that are not overlapped at the 28 days or after. Also, in Figure 4 the graphs are grouped as a&d, b&e, and c&f. The manuscript would be more accessible for the reader if the graphs of Figure 4 were grouped as a&b, c&d,, e&f.

-          Line 385: The indication of “Sensing” in the Figure 5a is not clearly explained in the text.

-          Line 389: In line 372 it is mentioned that “it was possible to detect three plots with low EC among the plots with 25% basal fertilizer (Figure 6)”. Further, explanation is required for this finding.

-          Line 515 and 516: “The rosette stage 2 to 3 weeks after transplanting was considered a suitable time for the survey” this is an assumption that you have made for your work, and it would be much more suitable for the conclusions section if you could justify its convenience based on the results that you have been able to gather throughout the course of your investigation.

-          Lines 521 and 522: As in the last paragraph of the introduction, please specify clearly between brackets which are the diagnostic methods that you have tested.

Reviewer 2 Report

Although, the authors came up with some interesting findings, title and abstract needs thorough revision based on the major findings. The introduction does not provide sufficient background. Though, the research design is acceptable and methods are adequately described, the results are not clearly interpreted and the discussion is also poorly written and seems insufficient. The whole manuscript should be revised carefully to improve English language and remove redundancies.

Reviewer 3 Report

In my opinion the article presented for review “  Applicability of variable rate nitrogen top-dressing based on measurement within-field variability of soil nutrients for cabbage production is characterized by a wide range of analysis.

The introduction is comprehensive and introduces well the research topic. Literature support with publications from the latest reports (in introduction).

However, remarks to:

Materials and Methods

-    Authors describe the methods of the experiment very scrupulously, but how long was the experiment? one year ? two years? It would be good to add information about the duration of the experience. You can add this information in the place where You write about where the experiment is conducted. I think one-year field experience after all, even if there is number of repetitions; it is not entirely representative.

-    From the presented diagram – fig 1 - I understand that each object had 3 repetitions? -this is not explicitly stated in the text
-    Fig 1 and the values…. 25%+ 50% etc, no explanation anywhere.
-    Fig 1- what mean cont?
-    The scheme of the experiment in the form of fig 1 is good, but it should be clarified with a legend. Which is trial 1 which is trial 2 and 3?
-    no information in the methods, about the number of repetitions of measurements- this information is in result. I think this information can be transferred to methods
-    Why table 4 is in discussion?

Results presented in a clear authoritative reliable form.
Presented conclusions are the result of the analysis carried out
Please read the MS  once more and correct any minor shortcomings, e.g. punctuation, etc.

Round 2

Reviewer 1 Report

Dear authors,

Thank you very much for your work. I consider that you have correctly addressed all my comments. If I may, I would like to suggest to move the explanation of the calculation of the cumulative temperature “Cumulative temperature is the sum of the daily average air temperature from the transplanting date to the day before harvest” (Lines 257 – 258) to the material and methods section.

Author Response

Responses to Reviewer #1

Reviewer#1

 Dear authors,

 Thank you very much for your work. I consider that you have correctly addressed all my comments. If I may, I would like to suggest to move the explanation of the calculation of the cumulative temperature “Cumulative temperature is the sum of the daily average air temperature from the transplanting date to the day before harvest” (Lines 257 – 258) to the material and methods section.

Response: We appreciated you for the peer review again. We are thankful for the time and energy you expended.

Response-Lines 133-135: We moved the explanation to the material and methods section as you suggested. Thank you for your advice.

Reviewer 2 Report

This paper can be accepted. Thanks!

Author Response

Responses to Reviewer #2

Reviewer#2

This paper can be accepted. Thanks!

Response: We appreciated you for the peer review again. We are thankful for the time and energy you expended.

Reviewer 3 Report

I think the article has been corrected. The authors addressed all objections. It may be published in a revised form.

Author Response

Responses to Reviewer #3

Reviewer#3

I think the article has been corrected. The authors addressed all objections. It may be published in a revised form.

Response: We appreciated you for the peer review again. We are thankful for the time and energy you expended.